# Hydrogel-Based Biosensors for Effective Therapeutics

**DOI:** 10.3390/gels9070545

**Published:** 2023-07-04

**Authors:** Mohzibudin Z. Quazi, Jimin Hwang, Youngseo Song, Nokyoung Park

**Affiliations:** Department of Chemistry and the Natural Science Research Institute, Myongji University, 116 Myongji-ro, Yongin-si 17058, Gyeonggi-do, Republic of Korea

**Keywords:** DNA-based hydrogels, biosensors, stimuli-responsive, progesterone detection, mRNA quantification, sensors

## Abstract

Nanotechnology and polymer engineering are navigating toward new developments to control and overcome complex problems. In the last few decades, polymer engineering has received researchers’ attention and similarly, polymeric network-engineered structures have been vastly studied. Prior to therapeutic application, early and rapid detection analyses are critical. Therefore, developing hydrogel-based sensors to manage the acute expression of diseases and malignancies to devise therapeutic approaches demands advanced nanoengineering. However, nano-therapeutics have emerged as an alternative approach to tackling strenuous diseases. Similarly, sensing applications for multiple kinds of analytes in water-based environments and other media are gaining wide interest. It has also been observed that these functional roles can be used as alternative approaches to the detection of a wide range of biomolecules and pathogenic proteins. Moreover, hydrogels have emerged as a three-dimensional (3D) polymeric network that consists of hydrophilic natural or synthetic polymers with multidimensional dynamics. The resemblance of hydrogels to tissue structure makes them more unique to study inquisitively. Preceding studies have shown a vast spectrum of synthetic and natural polymer applications in the field of biotechnology and molecular diagnostics. This review explores recent studies on synthetic and natural polymers engineered hydrogel-based biosensors and their applications in multipurpose diagnostics and therapeutics. We review the latest studies on hydrogel-engineered biosensors, exclusively DNA-based and DNA hydrogel-fabricated biosensors.

## 1. Introduction

The detection systems in therapeutics were widely studied in the previous decade due to the necessity for the rapid and robust analysis of human pathology [1]. The rise in drug toxins and side effects that emerged from drug overdose were the critical elements analyzed [2]. Previous studies showed a vast and emerging need for biochemical detection methods [3,4,5,6]. The conventional methodologies in the early detection of biochemicals were studied using HPLC, PCR, GC, and GC-MS [7,8,9]. However, these conventional methodologies are not highly favorable since robust, affordable, and highly sensitive detection methods are lacking [10]. Moreover, the conventional methodology does not favor working consistently under versatile humid conditions and also requires complex equipment with experienced operators [11,12,13,14,15,16,17]. On the other hand, researchers in the early 1900s developed polymer network nanocomplexes for biomedical applications. The use of architecture to design a polymeric network confined to hydrophilic properties aims to provide a tissue-like structure for biomedical purposes. Wichterle et al., in the 1960s, designed a hydrophilic gel (hydrogel) for the first time with the motive of designing a 3D polymer network to utilize in the human body [18]. These immense water-rich bodies of hydrogels easily adapt to the microenvironment due to the similarities in tissue-like structure, and their use in a significant dynamic range is feasible, as seen in Figure 1 [19,20,21,22]. Furthermore, hydrogels have surfaced as excellent sensory systems due to their high biocompatibility and variability which eases the tuning of gel chemistry [23]. In particular, the changes in the physical properties of hydrogels, such as sol–gel transitions can result in excellent target analytes [24,25]. Hydrogels are well known for their easy modulation of physical properties which tend to make hydrogels responsive to elements such as external stimuli, pH, temperature, ionic strength, light, and sound [26]. Generally, hydrogels are classified into different categories such as natural polymers and synthetic polymers. Hydrogels are fabricated with a polymer network using natural or synthetic materials with a high degree of flexibility owing to their large water content [27,28]. Natural polymers include chitosan, alginate, dextran, and hyaluronic acids [29,30]. Synthetic polymers include polyethylene glycol (PEG), poly (N-isopropyl acrylamide) (PNIPAAm), and poly (2-hydroxyethyl methacrylate) (PHEMA) [31,32,33].

The unique behavior of hydrogels in different physiological conditions can retain a large number of biological fluids. They are, therefore, ideal substance for a variety of applications. Hydrogels were closely studied based on the classification of their type and nature [27,34]. Subsequently, researchers developed a hybrid hydrogel with a mixture of natural and synthetic polymers to exert and utilize different elements’ unique properties [35,36]. Recently, the global pandemic showed the critical urgency and need for rapid, bio-susceptible sensors for the detection of contagious diseases and to deter their transmission [37]. Similarly, robust biosensor detecting systems were successfully utilized to examine biochemicals, chemical drugs, or toxins. These biosensors were utilized in a wide range of applications, including preventing and controlling drug abuse, preventing food contamination, and maintaining drug doses in body fluids to avoid drug overdoses [38,39,40]. In recent times several studies have shown a great overview of hydrogel-based optical ion sensors for point-of-care testing and environmental monitoring. Advanced applications in flexible wearable sensors, such as sweat sampling and flexible electrodes, were implemented. Moreover, several studies on the fabrication, application-based principles, and challenges in hydrogel-based biosensors have been well-reviewed [41,42,43,44,45,46]. However, our review focuses on the latest studies of hydrogel-based biosensors. Here, we explain their mechanisms of action, limitations, and future directions. We have briefly reviewed several first-time reported hydrogel-based biosensors with significant applications. We exclusively reviewed DNA hydrogel-engineered biosensors for their diversified role and limitations.

## 2. Hydrogels for Progesterone Detection

Progesterone plays a major role in the female body, it is a steroid hormone produced in the body to regulate the menstrual process, also known as a primary biomarker for reproductive status and monitoring fatigue and irregular menstrual cycles. Progesterone has a significant role in early gestation and regulation of progesterone levels which ease the process of pregnancy of unknown location (PUL) [47]. Analogously, Chen et al. have designed a PEG hydrogel with the assimilation of quantum dots (QD) and polyhistidine-tagged transcription factor (SRTF1) as a bioreceptor and Cy5 fluorophore-labeled cognate DNA sequence for the first time. The author demonstrates Förster resonance energy transfer (FRET) between a fluorophore-labeled cognate DNA and the modified QDs. The presence of progesterone leads to a decrease in fluorescence intensity (Figure 2I,II). The study was carried out in a dose-dependent manner and the sensor performs with a limit of detection of 55 nM. This strategy demonstrated the potential of the novel genomics-to-sensor approach using allosteric transcription factors to overcome the insufficient adequate bio-recognition of target elements [48]. Similarly, Velayudham et al., designed a chitosan and hydroxyethyl cellulose hydrogel conjugated aptamer-based electrochemical biosensor to detect progesterone. Here, the author used gold nanocubes (AuNCs) which were self-assembled with thiol-Au chemistry. The aptamers were modified with thiol and specific for progesterone (P4). The functionalities and successful applicability of aptasensors were tested using blood samples spiked at different concentrations of progesterone P4 [49] (Figure 2III). 

Moreover, Casis and the group studied a new method based on molecularly imprinted hydrogels and demonstrated a novel module designed specifically to recognize progesterone. The author fabricated hydrogel films via the copolymerization of acrylic acid and ethylene glycol dimethacrylate with 2,2′-azobisisobutyronitrile as initiator. The approach includes the non-covalent imprinting method with the colloidal crystal template technique to produce membranes with pre-specified morphology. This study suggests the engineered system can be used repetitively with high reproducibility and robust quantification of target molecules in photonic films through spectroscopic techniques [50] (Figure 2IV). 

## 3. DNA-Based Hydrogels as Biosensors

DNA-based nanostructures became adaptable building blocks for the manufacturing of soft materials at nanoscale level [51]. The flexibility of conjugation with other biomolecules, and the structural and functional properties of DNA made it favorable for various biomedical applications [52,53]. The hydrogels formed by DNA are biocompatible, stable, tunable, and biologically versatile, thus, these have a wide range of promising applications in bioanalysis and biomedicine [54]. Due to their characteristics and use in biosensing, bioimaging, and therapeutics, DNA-based hydrogels have attracted a lot of attention in recent years [55,56,57,58]. Nano-engineered structures and devices built with DNA nanotechnology have shown potential in a wide range of soft materials. Ned Seeman proposed DNA structure modulation at the nanoscale based on its physical properties. DNA is well recognized for its role as a genetic material, which emerged as an ideal building block at the nanoscale [59]. Next, the renowned Watson–Crick base pairing eases our understanding of it as a deterministic and controllable, sequence-specific hybridization between strands [60]. In particular, DNA nanotechnology has evolved in a transition from one-dimensional structures to three-dimensional structures. There are frequent studies on structure building in dimensions from DNA wires to nanotubes, lattice-crystal structures to diverse DNA polyhedra origamis with 3D structures [61,62]. To our knowledge and findings, DNA hydrogel’s role in biosensors is highly limited and rare to observe. However, in this section, we review several reported studies on DNA hydrogel-based biosensors in the sensing of microRNA, colorimetric sensing, point-of-care devices, and label-free detection-based applications. Additionally, we have overviewed the recent and well-approached unique terms such as “wireless infection detection on wounds” (WINDOW) with miscellaneous applications.

### 3.1. DNA-Based Hydrogels for Micro RNA (miRNA) Quantification/Detection

Recently, miRNAs have gained recognition as promising molecular biomarkers [63]. Earlier reported studies demonstrate several methods established to detect miRNA expressions, such as stem-loop reverse transcription PCR, rolling circle amplification, ligase chain reaction, and many other enzyme-catalyzed reaction-based techniques [64,65,66,67]. Moreover, abnormal miRNA expressions have been used to categorize, diagnose, and predict the prognosis of cancers. miRNA expression levels that are abnormally high are linked to several human diseases [68,69,70,71]. Therefore, researchers amend their ideas to construct hydrogel-engineered sensors for miRNA detection and quantifications. Hui Wang et al. developed a simply prepared capillarity self-driven DNA hydrogel sensor by fixing DNA hydrogel film at a capillary end. The direct hybridization between miRNAs and DNA probes from the hydrogel sense and detect miRNA targets. The DNA hydrogel sensor applies miR-21 as a proof-of-concept target (Figure 3I) [72]. Here, the authors performed a practical detection of miR-21 in the total RNA sample extracted from the MCF-7 cells. The recovery tests showcased that the miR-21 in the spiked sample was 54 fmol with a recovery of 107.5%. This visually quantitative detection of miRNAs stands, cheap and self-driven by DNA hydrogel sensors, along with the little volume of DNA hydrogel can detect targets successfully. Similarly in another study, Li and colleagues designed a versatile fluorescence strategy based on DNA hydrogels to detect microRNA-141s. A poly-directional hybridization chain reaction basing DNA hydrogel on SiO_2_ microspheres was designed. By coupling with DNA walking amplification, it served as a flexible fluorescence signal amplifier for the ultrasensitive detection of miRNA-141. The author claims based on the findings that this strategy can be applied to miRNA-141 detection in humans, especially prostate cancer with favorable precision, suggesting promising applications of the sensing strategy in disease diagnosis and biomedical analysis. The design and use of functional nanomaterials properties in sensing, detection, and imaging are widely known [73]. Li and group studied the selectivity of the DNA hydrogel-based fluorescence strategy for miRNA assay with several other miRNAs including miRNA-21, miRNA-155, miRNA-182, and mismatched miRNA-141 [74]. Based on miRNA responsive/targeting strategies Si et al. designed a novel surface-enhanced Raman scattering sensor array (SERS) with a Raman signal “ON” and “OFF” strategy. The design includes nine sensor units that can detect multiple cancer-responsive miRNAs in one sample. The researcher first synthesized DNA hydrogels followed by AuAg nanoparticles as SERS tags and incorporated certain MNAzymes (Multi-component nucleic acid enzymes) on a SERS array (Figure 3II) [75]. The design is based on the mechanism of miRNA interactions and AuAg alloy nanoparticles penetration of the hydrogels, which allow and interact with SERS arrays to produce Raman intense signals. The authors demonstrate a detection limit of 0.11 nM which was estimated using the 3σ rule. The miRNA interaction proceeds to configure and restore the activities of MNAzymes which lead to the breakage of the substrate linker, followed by the penetration of alloy nanoparticles on the hydrogel.

### 3.2. DNA-Based Hydrogels for Colorimetric Sensing Application

In the development of DNA hydrogel-based biosensors, cost-effectiveness, quick response, and ease of fabrication with handling are major concerns [76]. Colorimetric sensing has played a major role in addressing these concerns. Similarly, DNA hydrogel’s contributions towards similar issues were long standing. Several reports have demonstrated the functionalities of DNA hydrogel in colorimetric sensing. Here, in this section, we have reviewed the recent studies on DNA-based hydrogel for their application and significant role in colorimetric sensing. Earlier, Baeissa and group studied and engineered DNA-functionalized monolithic hydrogels [77]. The author utilized well-known thiol-Au chemistry to establish DNA-modified AuNPs and load them on DNA-modified polyacrylamide hydrogels. In the presence of the target, DNAs AuNPs adherence facilitates the visual detection regardless of instrumental analysis. The favorable point of this study was at even ~0.1 nM target where DNA can be visually detected by DNA-functionalized AuNPs. Moreover, this approach is easy to handle similarly to other homogeneous assays [78,79]. Moreover, recently, Zhao et al. reported a new type of DNAzyme-crosslinked hydrogel that contributes significantly to the rapid colorimetric sensing of H_2_O_2_ with the signal accumulation strategy [80]. Here, the authors used a G-quadruplex/hemin complex to catalyze the H_2_O_2_-mediated oxidation of 3,3′,5,5′-tetramethylbenzidine (TMB) to form blue oxidized 3,3′,5,5′-tetramethylbenzidine (oxTMB). The DNAzymes were assembled within the hydrogel network as cross-linkers, which played a major role within the gel to reduce H_2_O_2_ (Figure 4I). These peroxidase activities resulted in a color change of TMB. Similarly, the amine group influenced the electrostatic force to interact with the DNAzyme backbones to accumulate them. The author claims that this novel hybrid DNA hydrogel utilization in colorimetric sensing could be the first studied. Also, the DNAzyme-crosslinked hydrogels are regenerable and could be employed in the detection of H_2_O_2_ in environmental and food-related works with a detection limit of 1.0 µM of H_2_O_2_. 

Nanomaterial’s intrinsic properties have drawn major attention and played a significant role in designing biosensors [5,81,82]. Nanomaterials extinction coefficients, plasmonic properties, and the surface-to-volume ratio have facilitated new designs for the development of biosensors [83,84]. Recently, Liu et al. have designed a simple layer-by-layer assembly technique and developed a hybrid DNA-AuNP hydrogel film as a colorimetric biosensing system. The authors worked on DNA-AuNP hybrid hydrogel system to design a portable and storable biosensing system. The simple layer-based assembly structure was composed of densely packed AuNPs and crosslinked by DNA structures based on 3D hydrophilic networks which were well designed on a glass substrate (Figure 4II) [36].

The selective ion address caused DNAzyme activation and led to the degradation of DNA-AuNP hydrogel film. The ion-selective active degradation caused the release of AuNPs which further acted as sensitive colorimetric signals. These signals were read with a detection limit of 2.6 nM (Pb^2+^) and 10.3 nM (UO_2_^2+^). The designed biosensing system was significantly used in the colorimetric detection of Pb^2+^ or UO_2_^2+^. The DNA-AuNPs hydrogel-film-based biosensing system is highly promising for future application in rapid on-site detections.

### 3.3. DNA-Based Hydrogels for Point-of-Care Application

In recent years, researchers studied certain elements extensively to discover, ameliorate, and evolve the term “point of care” [41]. The evolution of point-of-care testing involves requirements such as stability, portability, ease of storage, and cost-effectiveness. Moreover, recently, DNA hydrogels have evolved as ideal signal transduction strategies for point-of-care functional roles. Jiang et al. proposed a facile DNA-based hydrogel capillary sensor (DHCS) for the sensitive detection of Pb^2+^. In this study, authors have proposed a strategy that inculcates a miniature, portable, sensitive, and highly selective visual platform for the detection of Pb^2+^ in situ measurement. The biosensors’ function demonstrated that in the presence of Pb^2+^, the crosslinker substrate strands were parted, which led the hydrogel to partially fracture. This strategy resulted in the solution flowing through capillaries, controlled by the hydrogel film which was affected by the concentration of Pb^2+^. Hence, the quantitative detection of Pb^2+^ was achieved regardless of instrumental analysis and observed with the naked eye owing to the distance and time delay in the process. Pb^2+^ as low as 10 nM could be directly detected by the naked eye. The author claims the sensor could be utilize to detect Pb^2+^ in tap water with significant reliability (Figure 5) [85]. Earlier, Zhu and the group demonstrated an aptamer-based target-responsive gel to detect non-glucose targets such as cocaine. The author designed a linear polyacrylamide with complementary aptamers and grafted two short DNAs. The study was directly proportional to the target concentration and engineered hydrogel. The work of action explains that glucoamylase was trapped inside the gel and separated its substrate on the outside of the gel (amylose), which further resulted in the gel’s collapse due to the target’s adherence to the aptamers. Later, Zhu et al. discovered the limitations of engineered hydrogel. The incorporation of gel into their earlier hypothesis demonstrates a novel quantitative assay to address the limitations of engineering an Au core/Pt shell nanoparticle (Au@PtNPs). The mechanism demonstrates that by enforcing negative pressure, the gel fractures and releases (Au@PtNPs) and, due to the force, the supernatant comes in contact with H_2_O_2_, which further decomposes into O_2_. The O_2_ bubbles propelled red ink into the top channel, and the ink’s migration distance was proportional to the target concentration. A hydrogel-entrapped Au core/Pt shell nanoparticle (Au@PtNP) and a volumetric bar-chart chip (HV-Chip) were used to visualize and quantify the data [86]. 

### 3.4. DNA-Based Hydrogels for Label-Free Detection

Optical biosensors were used in label-free technology, known as label-free detection, to measure the changes which take place after an analyte binds to a ligand immobilized on a biosensor surface [87]. The increase in global water pollution, industrial toxic waste/sewage discharge, and toxic heavy metal ions have become the main micropollutants globally. These methodologies raised concerns worldwide to address the rise in global pollution and engineer countermeasures worldwide. Analogously, label-free detection of such toxic metal ions studies have become appealing [88,89,90,91]. A strategical counter based on the new label-free method for Pb^2+^ biosensing was developed by Chu and colleagues, using a one-step preparation of Pb^2+^-responsive pure DNA hydrogel material. The substrate strand and Pb^2+^-dependent DNAzyme strand were added to fabricate DNA hydrogel. The work demonstrated that DNA hydrogel structures are destroyed when Pb^2+^ is present in the sample. The presence of Pb^2+^ activates the enzyme strand in the hydrogel skeleton which causes the substrate to be cleaved. With a minimum detection limit of 7.7 nM, DNA fragments released by the hydrogel’s collapse were easily measured as a signal output for quantifying Pb^2+^ concentrations. The author claims that this strategy could be applied to the field detection of toxic metal ions by modifying the DNAzyme and substrate sequences. (Figure 6I,II) [92].

The use of SPR-based techniques in biosensors is well known [93]. A similar approach with an enzyme-free and label-free surface plasmon resonance (SPR) biosensing strategy was designed by Guo et al. The authors engineered a DNA self-assembly aptamer-based hydrogel with streptavidin (SA) encapsulation, followed by ML/RARα (promyelocytic leukemia, retinoic acid receptor alpha) targeting capture probes (Cp) immobilized on the chip-surface-engineered Cp-PML/RARα duplex. The hydrogel nanostructure was established on the gold surface by targets triggered through X-shaped polymer self-assemblies. Next, Streptavidin aptamer’s selective binding facilitated hydrogel encapsulations and designed a high molecular weight complex. The high molecular weight structures were bound to the gold surface which increased the SPR signals and provided ultrasensitive detection. Based on this, several other studies have reported similar findings. This approach can detect targets up to the range from 100 fM to 10 nM with high efficiency [94]. The study demonstrated the potential to utilize this technique in clinical diagnosis for gene fusions with high detection capability.

### 3.5. DNA-Based Hydrogels for Target Responsive Applications 

The engineering of biosensors modulated with the advanced approach followed by selective target responses is needed urgently and is a trending topic. Tan et al. designed a graphene oxide (GO) hydrogel as a fluorescent biosensor for the first time based on a target-responsive strategy to detect the presence of antibiotics. The author constructed a 3D macrostructure by co-crosslinking the GO sheets with adenosine and aptamer. The adenosine and aptamers were employed as the co-crosslinkers to rope the GO sheets. The method includes a simple gelation, immersion, and fluorescence determination process which led to the development of a fluorescent sensing platform via GO hydrogel. The functional role of engineered hydrogel included high mechanical and thermal stability. The promising role of the hydrogel had the lowest detection limit, i.e., a limit to quantitation (LOQ) of 25 g/L. The first fluorescent GO-based target responsive study showed significant sensing for oxytetracycline antibiotics. The authors hypothesize that this study could be broadened towards other biomolecules through slight modifications in hydrogels [95]. 

Next, Ma et al. studied and designed a target-responsive aptamer cross-linked hydrogel for the visual detection of glucose. The target aptamer and two short complementary DNA sequences were used as cross-linkers which were attached to a linear polyacrylamide chain to design the glucose-responsive hydrogel. The thiol-PEG-modified AuNPs were used as the output signal for obvious detection which were encapsulated in the hydrogel. Ma et al. demonstrated that glucose complex can detect glucose easily with the naked eye with a sensor detection limit of 0.44 mM using a UV Visible spectrophotometric analysis. The boronic acid derivatives (Shinkai’s receptor) played a major role in the complex which ropes the aptamer to deform the hydrogel and causes the release of AuNPs with an evident red color in the supernatant (Figure 7) [96]. A similar study based on target-responsive DNA hydrogel with a chemiluminescent biosensor was constructed to sense adenosine. The designed DNA hydrogel carries prominent selectivity with a high loading capacity. The designed AuNPs coating on HKUST-1 (Au@HKUST-1) showcased stronger peroxidase-like activity than the original HKUST-1 (HKUST-1 Cu-based MOFs which have peroxidase activity). The Au@HKUST-1 was employed as a regulated chemiluminescent signal amplifier. A blend of partially complementary, acrydite-modified single-strand DNA, acrydite-modified adenosine aptamer, and hemin aptamer designed the DNA hydrogel. The DNA hydrogel completely disintegrated as a result of the interaction between adenosine and an adenosine aptamer. Whereas, in the presence of only hemin, a G-quadruplex/hemin was formed and, as a result, the DNA hydrogel remained integrated. Lin et al. studied the release of Au@HKUST-1 and G-quadruplex/hemin in the chemiluminescent system, followed by DNA hydrogel disintegration. The dual signal amplification of chemiluminescent biosensors was enabled. The author demonstrated the adenosine target-responsive detection in urine [97].

As aforementioned, the pollutant and drawbacks happening through metal ion-induced, sewage discharge, and toxic side effects in water bodies. This received researcher’s attention to design and develop constructive actions. Huang et al. engineered a DNAzyme-based responsive smart hydrogel system for the portable and quick detection of Uranium (UO_2_^2+^). Huang and group studied the hydrogel-entrapped AuNPs role in the visual detection UO_2_^2+^. Huang et al. used DNA-grafted polyacrylamide chains to crosslink enzymes and substrate strands of a DNAzyme complex to design a DNA hydrogel [98]. The encasing of gold nanoparticles (AuNPs) in the DNAzyme-crosslinked hydrogel and colorimetric analysis of the concentration of UO_2_^2+^ was accomplished. The enzyme strand is inactive without UO_2_^2+^. The UO_2_^2+^ in the sample medium activates the enzyme strand and caused the substrate strand to separate from the enzyme strand, which resulted in lowering the density of crosslinkers and weakening of hydrogel. The weakening of hydrogel allows the encapsulated AuNPs to be released. The author demonstrates through this strategy that it is possible to visually detect UO_2_^2+^ in quantities as low as 100 nM. Also, the target-responsive hydrogel worked in naturally occurring water that had been injected with UO_2_^2+^. Moreover, the authors studied and quantitively detected UO_2_^2+^ through the earlier reported “volumetric bar-chart chip (V-Chip)” to avoid errors [98,99,100]. 

### 3.6. DNA-Based Hydrogels for Miscellaneous Application in Sensing

Hydrogels oftentimes appeared as a tremendous and biocompatible element in designing biosensor systems [26,101,102,103,104]. Correspondingly, the conjoint approach towards a combination of biosensors and wireless technology could possibly open up the ease in diagnosing and treating medical conditions away from clinical settings [105,106,107,108,109]. Similarly, the lack of monitoring technology capable of interfacing pathogenic bacteria detection, tissue regeneration, and wirelessly transmitting data makes wound infections a significant clinical challenge. However, prompt detection is essential for effective interventions to control and monitor wounds. The gradual monitoring of wound wetness and bacterial growth could provide significant ideas to prevent the unintentional growth of foreign elements. Referring to this type of monitoring, Xiong et al. proposed the term “wireless infection detection on wounds” (WINDOW). The study demonstrated a sensing technology which significantly detected bacteria virulence using a DNA hydrogel-based, wireless, and battery-free sensor (Figure 8). Here, the author designed a strategy to detect an enzyme secreted by Staphylococcus aureus and Pseudomonas aeruginosa and Streptococcus pyogenes. The enzyme, known as deoxyribonuclease (DNase), was detected by a customized DNA hydrogel-based sensor that provided a radio frequency followed by enzyme detection. The study incorporates animal model-based demonstrations with near-field communications that are detectable by dielectric changes through the selective detections of deoxyribonuclease in pathogenic bacteria [110].

## 4. Outlook and Future Directions

In this review, we presented detailed and recent studies on the development of natural and synthetic hydrogel evolution and procurement as biosensors. We tried to reflect on different approaches in building and utilization of hydrogel-based biosensors in our review. Certainly, a particular hydrogel cannot be employed to solve and detect every analytical problem. In order to improve and perhaps even advance the performance of biosensors, we reviewed diversified hydrogels engineering, synthesis, and application in a variety of ways. In this review, we aim to provide a simplified study to reach the basic goals of 3D hydrophilic, water-rich bodies’ amplified role in biomedical, label-free biosensing, target responsive, and point-of-care detection roles. We specifically demonstrated DNA hydrogels and DNA-based hydrogel’s roles in a diversified field that showcases DNA as a bio-susceptible, eco-friendly, easy-to-produce, and cost-effective element for the construction of biosensors. DNA hydrogels could have enormous applications due to their easy adaptability and unique features, which are not limited to genetic materials. This review explains that DNA hydrogel studies are widely known, although studies of its specific target detection, enzyme-mediated stability, and easy conjugations with metal ions are still lacking. DNA hydrogels could evolve as potential biosensor-building elements to exert multidimensional roles and functionalities. In this review, earlier, we discussed several strategies and DNA hydrogel-based studies which clearly demonstrate stringent methods, lack of regenerations, and off-target delivery issues. The major outcomes throughout the reports were the “reproducibility” and lack of ease in handling the methodologies. Numerous studies on the fabrication and utilization of biosensors have been studied with spectral signal readout in vitro and in vivo. These techniques have high sensitivity and good selectivity; however, the stability and reproducibility of biosensors are suboptimal. Furthermore, reproducibility and reuse with significant detection limits were absent. We believe our review article could ease the understanding of biosensors, their designs, and their limitations. DNA hydrogels and also hybrid hydrogels could be engineered with the incorporation of selective DNA oligomers with ligands, carbon/QDs, or aptamers. This could advance the selective and sensitive detection and quantification of targeted analytes. Also, to design and construct the ideal biosensors, DNA hydrogels could be utilized as a potential building element/medium. The DNA nanotechnology fields are adapting and evolving. Therefore, future outcomes can focus on robust and selective target-based biosensors with great stabilities that can be conceived via DNA hydrogels at significant potential scales. The alternative strategies may include a complex of functional nanomaterials and hydrogels, enhancing the optical, electronic, and other physical properties. The unique features of such (SPR) surface plasmonic properties could be the one alternative method to extend the capabilities of such materials. Finally, in conclusion, as we reviewed, the outcome demonstrates that, to date, clinical trials and commercial-scale hydrogel-based sensors are not available. The recent COVID-19 pandemic and cancer survey by ICRA (International Agency for Research on Cancer) demonstrates the need for sensors in early diagnostics. In summary, the sheer need for advancement in diagnostics and early understanding of diseases could possibly reduce the hazardous impact of pathogen-caused diseases and malignancies. Also, the potential exposure of biosensors in clinical practices is still undermined. However, as aforementioned, recent research has shown that DNA hydrogel-based biosensors have a lot of potential as an alternative to traditional methodologies. The significant advantages of these tools are the broad range of applications and the adaptability of the sensing platforms. Hence, further investigation and research could possibly explore the regular employment of biosensors in clinical diagnosis.

## Figures and Tables

**Figure 1 gels-09-00545-f001:**
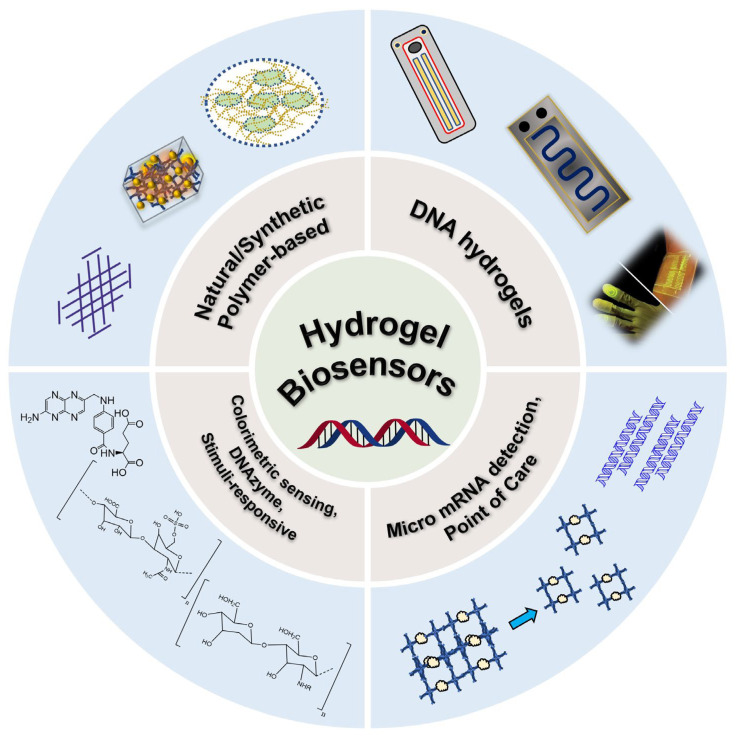
Schematic illustration of the designs of and functional roles played by hydrogel biosensors.

**Figure 2 gels-09-00545-f002:**
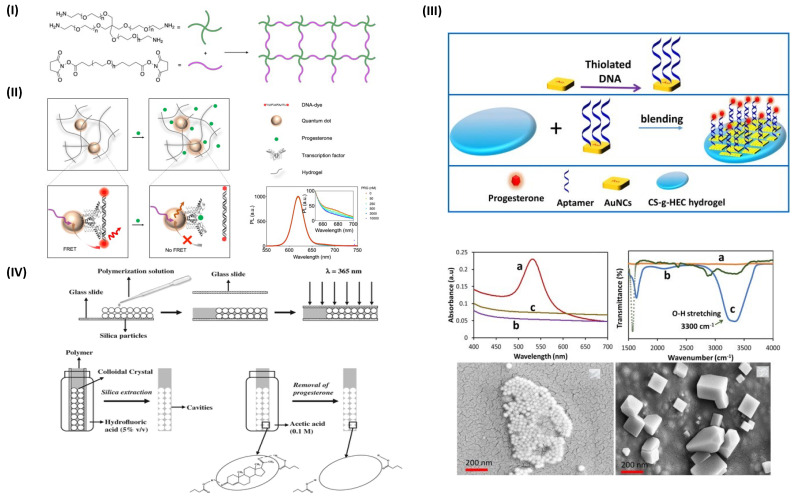
(**I**) Hydrogel precursors, formation, and swelling: 4–arm–PEG–NH2 and NHS–PEG–NHS. Hydrogel formation proceeds via a reaction between NHS ester (N-hydroxysuccinimide ester) and a primary amine, producing a 3D network. (**II**) Schematic illustration of the progesterone diffusion into hydrogel. A FRET–based sensor utilizing TF-DNA binding mechanism and signal attenuation in graphical observations. (**III**) Schematic representation of P4 aptasensor fabrication. Instrumental analysis of AuNCs (a), AuNCs-aptamer (b), and (c) chitosan hydroxyethyl cellulose hybrid hydrogel CS–g–HEC conjugated AuNCs-aptamer, FT–IR spectra of CS (a), HEC (b), and CS g–HEC (c); FE–SEM images of AuNCs and CS–g–HEC conjugated AuNCs-aptamer. (**IV**) Infiltration and polymerization process. Silica extraction and removal of progesterone. (Adapted from [48,49,50]).

**Figure 3 gels-09-00545-f003:**
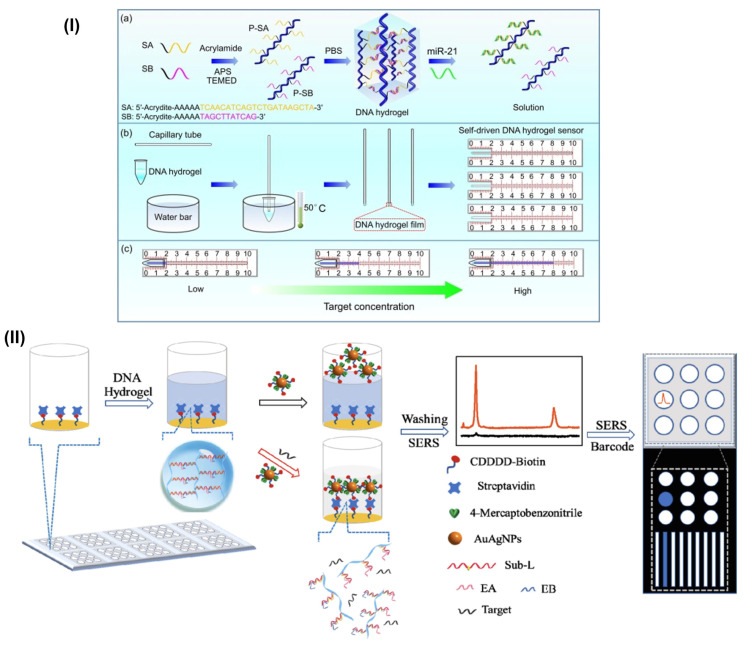
(**I**) (**a**) The designing/working principle of the self-driven DNA hydrogel sensor. (**b**) The preparation process of the self-driven DNA hydrogel sensor. (**c**) The principle of the visual and quantitative detection of miRNA using the proposed sensor. (**II**) Schematic illustration of the preparation and application of the target miRNA-responsive DNA hydrogel-based SERS sensor array for measuring multiple miRNAs in one sample. (Adapted from ref. [72,75]).

**Figure 4 gels-09-00545-f004:**
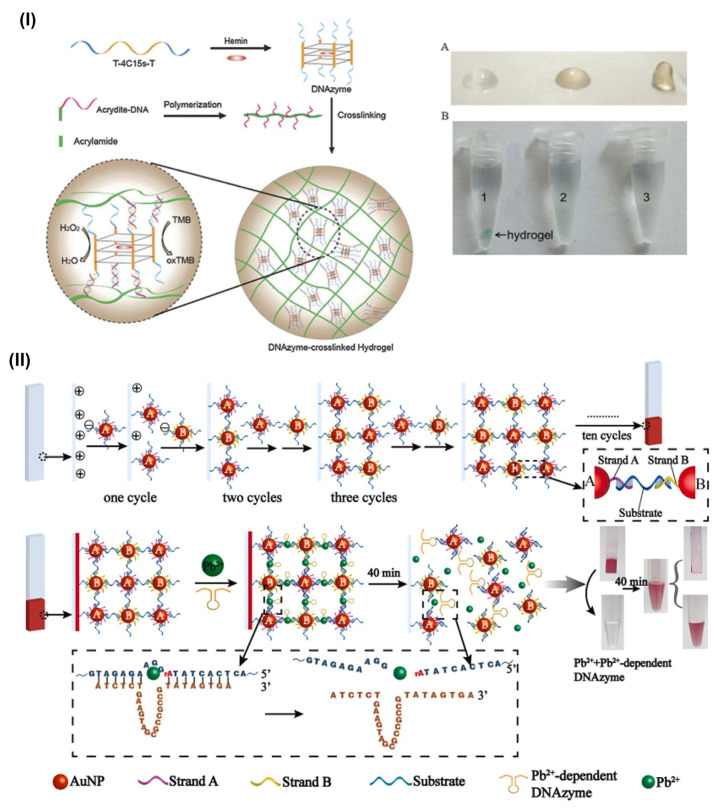
(**I**) Illustration of the preparation of the DNAzyme-crosslinked hydrogel. (**A**) Photograph of DNA-branched polyacrylamide solution (left), DNA-branched polyacrylamide solution containing 4c15s-DNAzyme without poly-T sequence (middle), and the T-4c15s-T-DNAzyme-crosslinked hydrogel (right) on parafilm. (**B**) Colorimetric sensing of H_2_O_2_ in the presence of DNAzyme-crosslinked hydrogel (1), free T-4c15s-T-DNAzyme (1.0 mM) (2), and hemin alone (1.2 mM) (3). Reaction conditions: H_2_O_2_ = 60 mM, MB = 0.2 mM, 20 1C, 50 min. (**II**) Schematic illustration of the preparation of DNA-AuNPs hybrid hydrogel film and the biosensing system for colorimetric detection of Pb^2+^. (Adapted from [36,80]).

**Figure 5 gels-09-00545-f005:**
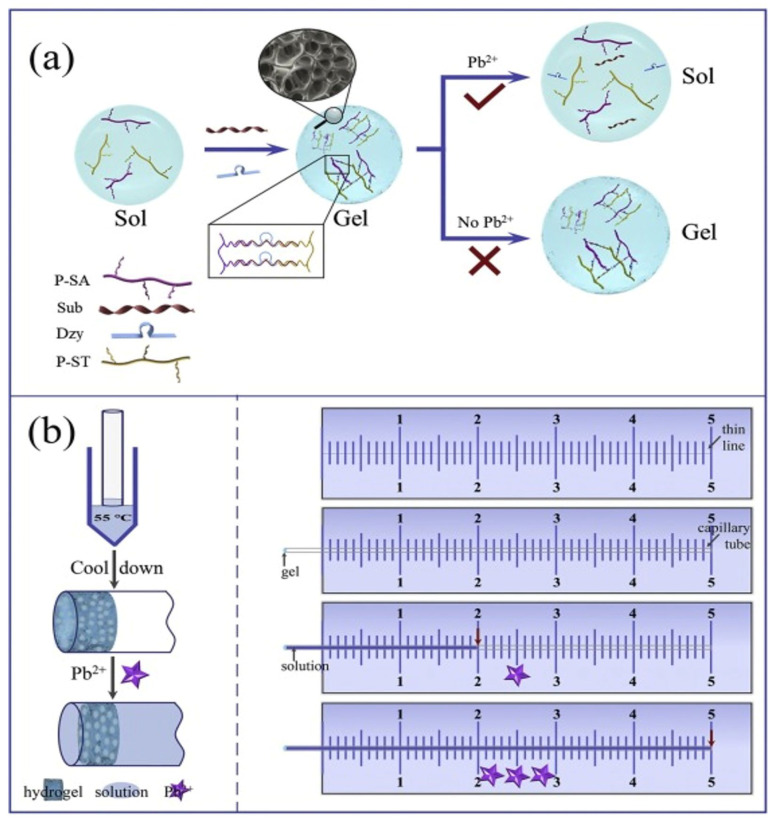
(**a**) Schematic illustration of the DNA-based hydrogel for the detection of Pb^2+^. (**b**) Fabrication and response processes are monitored by the DNA hydrogel capillary sensor (DHCS). (Adapted from [85]).

**Figure 6 gels-09-00545-f006:**
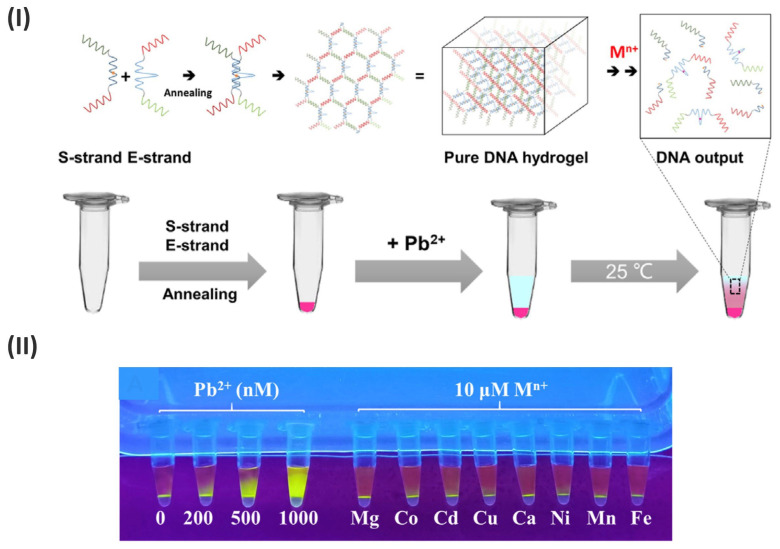
(**I**) Schematic of DNA hydrogel construction and metal ion response. Preparation of responsive pure DNA hydrogel and principle of metal ion detection. The E-strand and S-strand are mainly composed of complementary domains, from 5′ to 3′, respectively. DNAzyme sequence highly specific for metal ions such as Pb^2+^ was designed in the middle domain of E–strand. After the target is added to the hydrogel, it can gradually diffuse into the hydrogel and activate the DNAzyme to cleave the substrate, causing the breaking of the hydrogel. (**II**) The visual verification and selectivity of the responsive DNA hydrogel for Pb^2+^ (Adapted from [92]).

**Figure 7 gels-09-00545-f007:**
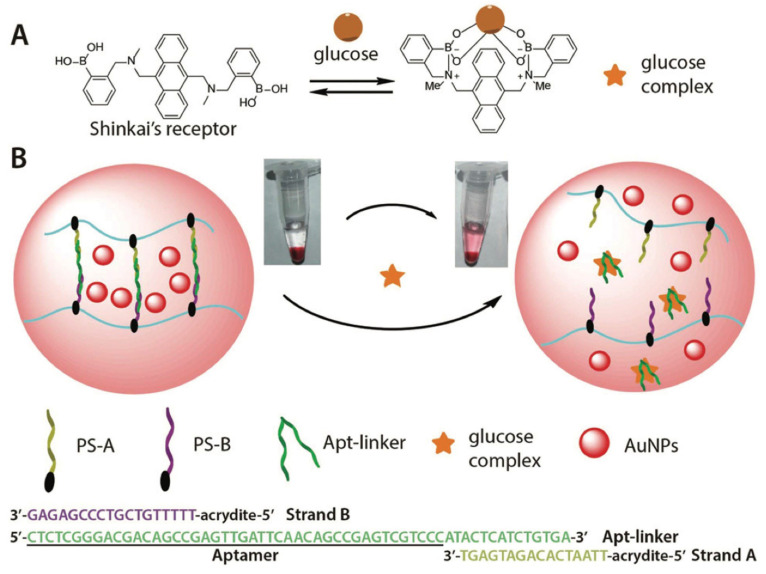
(**A**) Schematic illustration of the fabrication of glucose complex. (**B**) Function of the DNA hydrogel-embedded AuNPs for the visual detection of glucose. In the presence of the glucose complex, the DNA hydrogel is disrupted, and encapsulated AuNPs are released into the supernatant. The supernatant’s red color can be observed with the naked eye. (Adapted from [96]).

**Figure 8 gels-09-00545-f008:**
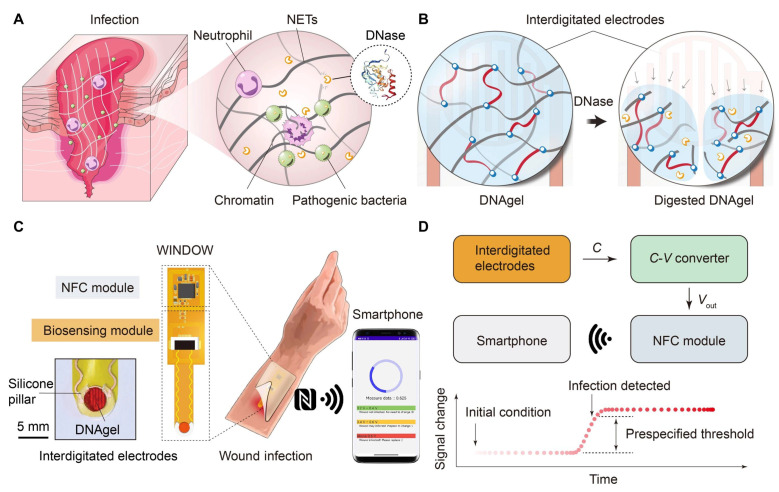
(**A**) DNase is a virulence factor in wound infections. Pathogenic bacteria secrete DNase to evade neutrophil extracellular traps (NETs), which are integral to the host’s immune response. (**B**) Schematic of the infection sensing mechanism. DNA gel is degraded upon exposure to DNase, resulting in a change in the capacitance of the sensor. (**C**) Schematic of the wireless wound infection sensor. WINDOW integrates the bio-responsive DNA gel, a half-wave–rectified LC biosensing module, and an NFC module to enable smartphone readout of the wound status. Inset image: Sensor-integrated DNA gel stained with rhodamine B. (**D**) System block diagram showing signal transduction from the DNA gel-based biosensor to the NFC module and to a smartphone for wireless readout and display. (Adapted from [110]).

## Data Availability

The data presented in this study are available upon request from the corresponding author. The data are not publicly available due to Ethical considerations.

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
