# Peer review of "Hydrogel-Based Biosensors for Effective Therapeutics"

_gels, 2023, doi:10.3390/gels9070545_

Round 1
Reviewer 1 Report
This is an excellent work, highly important and timely review of principles and applications of DNA-hydrogels in biosensing. I particularly praise mentioning of further development of an old idea of molecular imprinted polymers based sensors functionalised with DNA hydrogels to increase the specificity of detection. Applications of DNA-hydrogels in wound dressing is very promising too.
The review can be published in the present form.
Reviewer 2 Report
The article provides a large list of references and describes many applications for hydrogel sensors. The content, however, lacks structure and added value: The title is too general and does not indicate that in fact only hydrogel sensors are covered. It should be rewritten.
The sectioning is not fully comprehensible. Why is section 1.1. on hydrogel sensors for progesterone detection part of the introduction whereas all other sections thereafter form part of main section 2? The introduction is also not helpful. There is a motivation on the relevance of the topic, but a quite short and general one. But there is no discussion on existing review papers on the topic. There is also no precise explanation of the actual goal of the paper.
In its current state, the contribution provides a broad and interesting list of resources and facts on sensors. For an added value, however, there should be a clear goal what this overview should actually provide and why researchers should read it. I had expected a better categorisation of the materials or principles. There is a categorisation with respect to applications, but it is still too weak. The sources and their description should also be more goal oriented. Now, it is a mere description with very little comparison between each other about their pros and cons.
The conclusion and outlook is also too general. There is hardly any discussion on the presented works and facts, which could lead to a short and concise conclusion. The outlook is a bit vague since it is not based on a clear and comprehensible conclusion.
The figures are nicely collated and brought into a form that they fit well together. But they also provide facts, of which the goal is not really clear. What should be proven, or how do these figures assist the discussion in the text?
Especially the abstract makes the impression of an automatically generated translation. There are some ill-formed sentences and adjectives that have slightly different meaning than the intended one in the text.
The latter occurs on a regular basis in the text.
Reviewer 3 Report
The review article by. Quaziet et al., "A Polymer Instigated Path in the Engineering of Sensors and Biosensors for Effective Amelioration of Therapeutics ". The authors discuss recent research in synthetic and natural polymer-engineered hydrogel biosensors and their applications in multipurpose diagnostics and therapeutics. The subject is interesting and valuable. However, there are some points which should be considered by the authors for further completion of the review, as commented below:
1. Abstract - Please provide a comprehensive abstract that covers the problem and the study's objective, as well as materials and methods, results, and conclusions. The abstract section should provide an exhaustive statement of the highlights of the paper and appropriately state the conclusions that have been accomplished. Please reshape it.
2. Introduction - Authors should be given about the previous related studies done. Also, provide detailed and informative information about published articles. At the end provide the importance of the study and objectives selected for the study. For example, how to functionalize hydrogels or the application fields of hydrogel engineering biosensors, etc.
3. The author briefly reviews several influential and important applications of hydrogel-based biosensors; quantitative data should be added to highlight the advantages of hydrogel biosensors. The introductory sentences need to be avoided. Please reshape it.
4. It is well known that hydrogels and hydrophilic polymer networks play important roles in biosensors and biomedical engineering due to their good biocompatibility, biodegradability, hydrophilicity, and mechanical properties. The chemical structure and response mechanism of the polymer should be included. In addition, the non-specific adsorption performance statement for real sample detection should be added.
5. In section 2.7 DNA Hydrogels induced miscellaneous application in sensing. A statement of relevant research in this area should be added, as it is the focus of current research in this area.
6. Conclusion - The authors conclude by writing that "The major outcomes throughout the reports were the reproducibility and lack of ease in handling the methodologies. Numerous studies on the fabrication and utilization of biosensors have been studied with spectral signal readout in in vitro and in vivo. These techniques have high sensitivity and good selectivity; however, the stability and reproducibility of biosensors are suboptimal. Similarly, reproducibility, and reuse with significant detection limits were absent". It is impossible to express the main point of this article. Conclusion should be enriched with an outlook devoted to challenges facing hydrogels biosensor and future development directions. Please reshape it.
Round 2
Reviewer 2 Report
The paper has slightly improved. But the main problems still persist. The sectioning is really confusing. There is no Section 2, but it starts with 2.1. Then, section 3 introduces DNA-based hydrogel sensors. But in the introduction and in the abstract it is mentioned that the whole article is about DNA-based sensors. The subsections in 3 do not have a convincing categorisation: Some subsections cover specific working principles, some others highlight application advantages, a last one focuses on other sensing applications.
In a review paper, the most important aspects are the listing of relevant Ressourcen (which seems to be the case). But at least as important is a good categorisation and grouping of the research in order to provide more insight and to shed light on trends. Therefore, the authors should think of the importance and the rôle of section 2.1: It might be skipped completely or introduced in a better way. My suggestion would be a section that explains the methodology (as requested from reviewer #3) how the sensors have been categorised and what the inner logic of the article is meant to be. Section 3 would be the actual overview. Maybe this could be divided into working principles in one section and application aspects in the other. The summary and outlook should give a good analysis of the trends that can be seen in the review paper and where the next open point are.
Therefore, the paper is still not suitable for publication and should be revised.
Some sections start with lower-case letters. Some sentences are hard to understand. But the overall quality of the article is good.
Round 3
Reviewer 2 Report
The numbering of the sections have been adjusted. But I still do not see the point in Section 2, whereas all other reviewed activities are about DNA-based hydrogels. I strongly suggest to skip this paragraph completely. But because my co-reviewers seem to be okay with the state of the manuscript, I will not obstruct the process and leave it to the authors to decide.
The contents of the reminder of the paper are good and can be left as they are. I strongly suggest, however, a more focused title of the manuscript. There is neither an indication on hydrogels nor on DNA-based ones.
Also, the titles of the sub sections are not really helpful and should be improved. Either stick to applications or to working principles/mechanisms.
